# Low Entropy Future Boundary Conditions

**DOI:** 10.3390/e24070976

**Published:** 2022-07-14

**Authors:** Lawrence S. Schulman

**Affiliations:** Physics Department, Clarkson University, Potsdam, NY 13699, USA; schulman137@gmail.com

**Keywords:** cosmology, nuclear physics, low entropy future boundary conditions

## Abstract

A number of ways to detect future, low-entropy, boundary conditions are considered. The most important of these is the use of slowly-decaying isotopes and the observation (or prediction) of galactic dynamics. There is the expectation that future developments in experimental or observational technique will yield positive results.

## 1. Introduction

This article is an attempt to see whether one can detect, observationally or experimentally, whether there are future boundary conditions. In particular we are concerned with future boundary conditions that involve low entropy. The conclusion is “probably not at this time”, but there is reason to try. The first reason is that one or two of the methods described here may soon succeed. Another is simple curiosity, but it is also true that there is a quantum measurement theory [1,2,3,4] which is difficult to justify in any other way.

It is clear that, if things have come to equilibrium between initial and final boundary conditions, nothing will be evident; see Figure 1. It is just as clear (from the figure) that a system that does not come to equilibrium is needed to see future boundary conditions (for a fuller explanation of the figure, see Appendix A.)

Wheeler was aware of this problem, and proposed that long-lived nuclear isotopes might be an answer [5]. Unfortunately, he thought this could be done in the laboratory (and see a difference between an exponential and a hyperbolic cosine). At that time I wrote a paper [6] (see Appendix B) pointing out that you would require knowledge of the total abundance of the substance in the universe, presumably impossible to get.

All this would seem like idle speculation if, as now seems to be the case, the universe will go on expanding forever. However, doubts have been raised and there may be low entropy conditions in the distant future [7,8,9]. True, there is neither experimental nor observational evidence in favor of a “big bounce” or similar events, but one should bear in mind that this would not be the first time that there have been surprises in cosmology.

It is therefore relevant to ask: what are the slow relaxation times? We take up a number of slow processes.

## 2. Various Indicators

*Observations and N-body results:* Dynamics on a cosmological scale are certainly slow, a condition for seeing future boundary conditions (cf. Figure 1). One could make the case for low-entropy future boundary conditions if one could see that galaxies (as points) were not equilibrating. The points would still follow the equations of motion, and it would just appear to be unusual behavior. However, as far as I know there is no such indication. (They seem to virialize quickly [10], but presumably that is not related.) There are two possibilities. The first is *N*-body simulations. However, simulations are not sufficiently accurate to answer this question. After all, there are uncertainties connected with dark matter as well as other sources of possible error. Another problem is that the non-thermodynamic behavior might be so far into the future as to be impossible to see. This deals with an issue in any theory that purports to establish future boundary conditions: how far into the future are we talking about?

But there are other data. The range of fractality is subject to dispute as well as the definition of the range of inhomogeneity (and they are closely related [11]). At the time of recombination the range of inhomogeneity was small, near zero. It is now large, but is it infinite? This is where a dispute occurs [12], but the consensus is that it is finite [13]. How large depends on your definition and what era you are looking at.

Graphs of the range of inhomogeneity are given in [14,15,16] and in Figure 2 (see also [17,18,19]). There is little indication of a decline at z=0, that is, at present. It could be that the first signs of this slowdown may be in the distant future—at most this can give a bound. It is possible, however, that with greater precision it would be possible to see this effect.

*Dirac’s theory:* The classical theory of the electron’s radiation involves future boundary conditions [20], and in fact Dirac calls this (on page 157 of the cited paper) “the most beautiful feature of the theory.” Radiation is a damping process and Dirac ends up with odd-order derivatives having definite signs. In order to suppress runaway solutions he must invoke future boundary conditions. However, this has particular consequences: there is pre-acceleration. In other words, the electron “knows” it is about to be accelerated and undergoes some degree of acceleration *prior* to the forces that cause it to be accelerated.

There are two defects in looking for this effect. Firstly, the theory is classical. Secondly, the time of “pre-acceleration” is extremely short, on the order of $m/e^{2}$, which for an electron corresponds to about $6\times 10^{-24}\;$ s (it is actually $\frac{2}{3}\frac{e^{2}}{mc^{3}}$ that is $6\times 10^{-24}\;$ s.)

This is not an example of reduced entropy, but is included to disturb those who see no need for future boundary conditions.

*Nuclei*: There are a number of long-lived isotopes in the half life range of $10^{15}$ to $10^{24}$ years; see Table 1. Can these be used to indicate that there are future boundary conditions?

As indicated, you would need to know global abundance of an isotope. All those in the table decay by rare modes, the double electron mode being the longest lived. Recently, the double electron mode of ${}^{124}$Xe was observed [21,22], and is the longest lifetime actually measured. There is little information, however, on the overall abundance of some of these isotopes. Even what is available applies to the solar system or at most to our galaxy. However, this is an active area of research [23,24,25,26,27,28,29] and, for our galaxy, some of the isotopes in our table will have their abundance known. If we can assume that our galaxy is typical (or the relative production known) then the creation rate of those isotopes would also be known.

For example, Alibés et al. [23] discuss all elements up to the peak in iron (which includes V and Ca), in particular the abundance in the galaxy. The s-process deals with the heavier elements. There were indications in the work of Srinivasan et al. [30] (on the Murchison meteorite) that “stardust”, could be an important factor. Bisterzo et al. [24] are concerned with the flux of neutrons and what effect that would have on isotopic composition. Another factor is the effect of cosmic rays, and that is studied in Cook et al. [25]—there is an increase in tungsten-180 that concerns them. More specific information is sought by Den Hartog et al. [26], in that the paper deals with the abundance of elements in the sun and metal poor stars. Peek [27] is concerned with europium and reaches conclusions about the overall frequency, although the vagaries of data availability induce a confinement to stars suspected of having exoplanets. Roederer et al. [28] have an interest in r-process tellurium; again they look at metal poor stars, but this gives an idea of galactic abundance. Finally (in my list of citations), Travaglio et al. [29] see the galactic abundance of Sr, Y and Zr, so the question revolves (as it does for most contributors) around whether our galaxy is typical or at what stage it is. It should also be realized that my list of citations is not complete. Moreover, although this work is at most galactic in nature, the production rate for our galaxy is known for some elements. Moreover, the place of our galaxy among those that exist or existed may be guessed at with some degree of confidence.

A problem is that, even if long times are involved, it is difficult to distinguish exponential decay from the hyperbolic cosine (as predicted by Wheeler and verified *in some cases* by me [5,6,31]). The quantity $\exp\left(-x\right)-\cosh\left(X_{\max}-x\right)/\cosh\left(X_{\max}\right)$ differs from zero by less than 10${}^{-2}$ for $X_{\max}=5$ (and x≥0). Furthermore, this assumes there is a definite value for the decay rate. For suppose there was a hyperbolic cosine that dominated. Wouldn’t that be interpreted as a slower decay?

On the other hand, with rare isotopes continuously being created, a simple hyperbolic cosine instead of an exponential would not be the dependence. However, given good (maybe unrealistic) control over global decay rates, there may be a way. As discussed in [6,31], in a small environment, for example a laboratory where the lifetime is measured, the fundamental lifetime would be observed. However, when looking at global events, the lifetime would be adjusted so that final conditions would be observed, those final conditions having none of the long lived isotope present, so the observed lifetime would be shorter than the laboratory value. There are two difficulties. First to make sure it is global, and not confined to our galaxy, and second to determine the lifetime of the remote or galactic sample.

*Long lived substances between stars*: Diamonds are not forever, but they and other solids can last a long time [32]. In fact, if time is symmetric on some scale, one would expect “relics” from the future. There would be two requirements: first this relic must escape from the fiery holocaust of a sun exploding. Second, it would be necessary to distinguish these relics from forward-time objects.

The second requirement may be satisfied by showing extreme age, for example in the absence of some slowly decaying isotope. A gold ring might have uranium as an impurity. Within $1.4\times 10^{10}$ years it will decay, so gold without uranium (and lead instead) is old. Of course, techniques of purification might eliminate the uranium (and hence the lead), but that too is reminiscent of the future. (To be precise, half the uranium will decay by that time, so for my conclusion a few half lives should have passed.)

Since I have no way to estimate this possibility, nor has any relic been found (although it may be sitting in a museum somewhere) it is speculative. However, should techniques of identification or estimation become available, this would represent another avenue.

## 3. Black Holes?

The formation of black holes would seem irreversible. Yes, there is black hole evaporation, but the time scale is long, orders of magnitude longer than, say, measured nuclear decays.

This means that if there are future boundary conditions and they force a reduction in entropy, than either they take place in the very distant future, or there is a flaw in the reasoning that leads one to believe in their effective irreversibility. The latter could easily occur, given the singularities that appear in those conclusions.

## 4. Doubts about Conventional Cosmology

The discovery of accelerated expansion [33] is a case in point. There have been doubts about this discovery [7,34,35,36,37,38,39,40,41,42,43]. These consist of under density of our local portion of the universe, leading to questions about the conclusion that there is expansion. Buchert [7,42,43] is particularly concerned with “backreaction”, meaning the nonlinear effect of ignoring all but averages of density. The questioning and doubts are fairly common in cosmology and leads to surprises (which should not really be surprises).

## 5. Conclusions

There is insufficient observable evidence to establish a final restrictive boundary condition. There is theoretical reason to entertain this idea, but theory is not enough. In most cases, fortunately, there is hope that improved techniques will ultimately yield a positive result.

The techniques that seem hopeful are (1) slowdown in levels of homogeneity growth and (2) comparison of remote decay rates compared to local ones. These (and their flaws) are summarized as follows.

In the case of galactic dynamics, the range of the beginnings of homogeneity are available. If that level shows signs of slowing down it would indicate a future low entropy boundary condition. However, the range is subject to controversy and I would expect a slowdown would become controversial if only because of the definition of “homogeneity”. However, controversies are not new to cosmology.

Nuclear physics also has problems. The unaffected decay rate may be measured. The controversy would arise when galactic or extra-galactic rates are deduced. I would expect disagreement on galactic rates, even if naively they look slower.

As indicated I expect objections to be raised. However, this will not prevent measurements and observations. Moreover, if there is an indication of a slowdown, it will be noted.

## Figures and Tables

**Figure 1 entropy-24-00976-f001:**
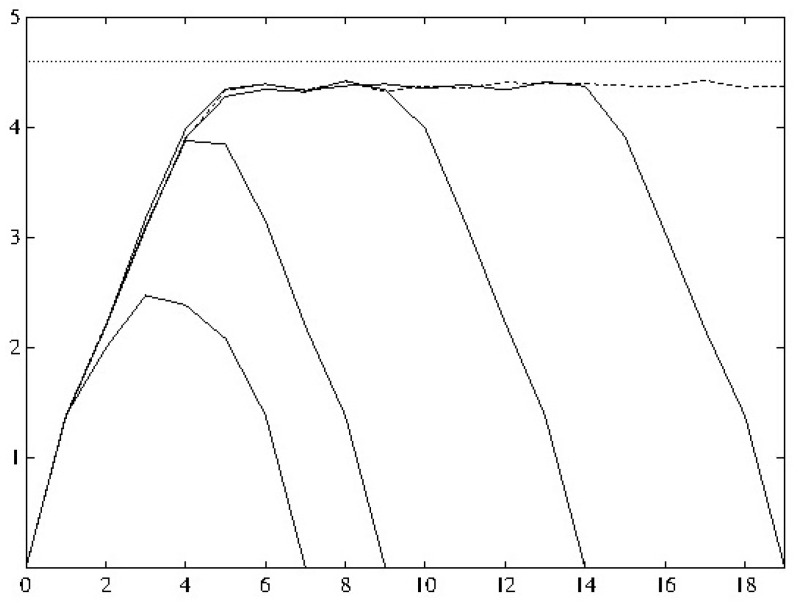
Entropy as a function of time for the “catmap”. (See Appendix A for further explanation.) There are various future boundary conditions including (dashed line) none. Those boundary conditions where equilibrium can be established cannot be distinguished from unconstrained evolution. However, when equilibrium cannot be established—e.g., boundary conditions at time-7—differences are apparent.

**Figure 2 entropy-24-00976-f002:**
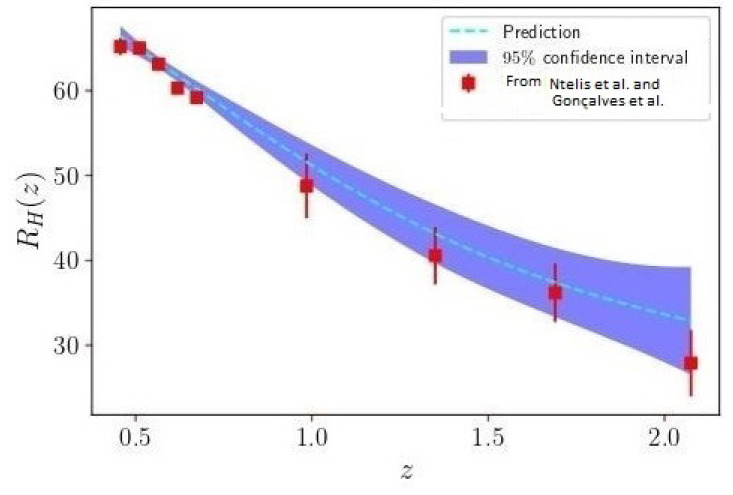
Reconstruction of the homogeneity scale function $R_{H}\left(z\right)$ (dashed line) and the $R_{H}$ measurements (red squares) presented in [15,16]. There is some hint of a slowdown at “recent” times but it is not significant, nor are definitions definitive. Adapted from [14], Figure 2.

**Table 1 entropy-24-00976-t001:** Isotopes with half-lives in excess of $10^{15}\;$years. For reference, the consensus on the time since the Big Bang is about $1.4\times 10^{10}\;$years.

Isotope	Half-Life (years)	Isotope	Half-Life (years)
hafnium-174	2.002 $\times \;10^{15}$	zirconium-96	20 $\times \;10^{18}$
osmium-186	2.002 $\times \;10^{15}$	bismuth-209	20.1 $\times \;10^{18}$
neodymium-144	2.292 $\times \;10^{15}$	calcium-48	23.01 $\times \;10^{18}$
samarium-148	7.005 $\times \;10^{15}$	cadmium-116	31.02 $\times \;10^{18}$
cadmium-113	7.7 $\times \;10^{15}$	selenium-82	110 $\times \;10^{18}$
vanadium-50	140 $\times \;10^{15}$	barium-130	1.2 $\times \;10^{21}$
tungsten-180	1.801 $\times \;10^{18}$	germanium-76	1.8 $\times \;10^{21}$
europium-151	5.004 $\times \;10^{18}$	xenon-136	2.165 $\times \;10^{21}$
molybdenum-100	7.804 $\times \;10^{18}$	krypton-78	9.2 $\times \;10^{21}$
neodymium-150	7.905 $\times \;10^{18}$	xenon-124	18 $\times \;10^{21}$
tellurium-130	8.806 $\times \;10^{18}$	tellurium-128	2.2 $\times \;10^{24}$

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
