# Peer review of "Low Entropy Future Boundary Conditions"

_entropy, 2022, doi:10.3390/e24070976_

Round 1

Reviewer 1 Report

This paper discusses a possibility to detect future, low entropy, boundary conditions of the Universe. This is a legitimate question in contemporary cosmological, namely in connection with such proposals as Penrose's conformal cyclic cosmology, Khoury, Steinhardt and Turok's ekpyrotic Universe, false vacuum catastrophe or a Big Crunch.

The paper is a nice compilation of author’s former results (some of them published in his book) that are logically knit together with some recent findings.

I have found all steps in the manuscript clearly stated and the logical thread of the argumentation possible to follow rather easily.  Overall this is an interesting paper that adds value to the literature on the chronology of the Universe.

Despite the fact that the manuscript is quite short without much supporting mathematics --- more like nice essay, I feel that it deserves to be published in Entropy in its present form.

Author Response

Thank you.

Reviewer 2 Report

The manuscript under review discusses possibilities to predict the so-called "future boundary conditions". A number of approaches are considered leaving the final prediction uncertain in the end.

The abstract and conclusions are too short and not very informative.

The manuscript is overall written in a rather obscure way and is not intended for a wide readership. In some places, the Author heavily relies on his previous results and this makes the manuscript not easy to follow and comprehend.

There are also several minor issues listed below.

Lines 29-30: "Dynamics on a cosmological scale is certainly slow, a needed condition (from the figure)." 

-- something must be missing in this sentence.

Page 2, the "f.equil" string in the caption of Figure 1 is strange; a LaTeX command might be missing 

Page 3, lines above and below Table 1 contain some strange chains of symbols (f.avila ... t.list), now I think that the \label{...} commands might be missing.

The numbering "Appendix F" looks strange. Perhaps, the section counter should be set to zero before the Appendices.

In references [1-4,5,8,14,16,19-21,etc.] journal titles are missing... If the reference list was generated via BibTeX, a systematic error in the .bib file might be to blame.

Author Response

The abstract and conclusions have been elaborated and now include more substance.

Also I have gone through the paper and have better explanations of what I am talking about. In particular the sentence near the beginning has been modified. But no substantive changes have been made. There is also more explanation of my previous work on the subject in App. B.

The latex issues are alarming to me, as I have no such issues in my pdf files. I have attached a pdf for this referee's convenience and have asked the editor about what causes the errors.

Round 2

Reviewer 2 Report

The changes made might warrant the publication